# Environmental conditions limit attractiveness of a complex sexual signal in the túngara frog

Wouter Halfwerk [1,2], Judith A.H. Smit[1], Hugo Loning[1], Amanda M. Lea[3], Inga Geipel[2], Jacintha Ellers[1] & Michael J. Ryan[2,3]

Animals choosing particular display sites often balance sexual and natural selection pressures. Here we assess how physical properties of display sites can alter this balance by influencing signal production and attractiveness of the túngara frog (*Physalaemus pustulosus*). Males that call from very shallow water bodies (few mm depth) benefit from reduced predation risk, but by manipulating water levels, we show that this comes at a cost of reduced attractiveness to females. Our data show that calling from shallower water reduces a male's ability to float, limits the inflation of his vocal sac, and consequently reduces signal conspicuousness in terms of amplitude and complexity. Our results demonstrate that display site properties can set limits on signal production and attractiveness and may hence influence signal evolution. Signallers may shift between sites or engineer their display location, which can play a crucial role in signal divergence and speciation, particularly in a rapidly changing world.

[1] Department of Ecological Sciences, VU University, Amsterdam 1081 HV, The Netherlands. [2] Smithsonian Tropical Research Institute, Apartado 0843-03092 Balboa, Republic of Panama. [3] Department of Integrative Biology, University of Texas, Austin, TX 78712, USA. Correspondence and requests for materials should be addressed to W.H. (email: w.h.halfwerk@vu.nl)

Sexual displays have evolved in response to selection imposed by mates and rivals and are generally assumed to enhance an individual's attractiveness[1, 2]. Animals can enhance their attractiveness by adjusting their signalling behaviour, such as increasing their display rates, by adding ornaments, or by improving signal conspicuousness[3–5]. Alternatively, animals can change the location from where they display to improve signal efficacy and consequently attractiveness. The habitat features of display locations are well-known to influence the efficacy of signal transmission and perception[6–10] and display site properties are often correlated to signal characteristics[11, 12]. Cichlids species displaying at different water depths in Lake Victoria, for example, show a spectral match between their body coloration and depth-dependent luminance[13]. Choosing a particular display site can thus enhance or reduce a signaller's attractiveness.

Signallers are often confronted with eavesdroppers, such as predators or parasites, which can also use signal components for detection and localisation[14–16]. The benefits of displaying from certain locations in terms of sexual attractiveness can therefore be balanced by costs associated with increased predation or parasitism[17]. Birds singing from treetops, for example, may improve the transmission of their song[18], but also risk more predation compared to birds singing from lower branches[17]. The choice of display site thus influences both natural and sexual selection pressures imposed on signals and can therefore play an important role in signal evolution.

Anuran vocal communication provides an excellent opportunity to study the link between display site and signal evolution. Male frogs call from a wide variety of microhabitats that can be situated on land, in water, or on vegetation, and their call characteristics can match display site properties, such as species calling from burrows[19–22]. The choice of display site is also known to be under strong natural selection from predation. In particular, many frog species call while floating on the water surface, making them easily detected by visual predators such as birds. Calling from the water surface also induces surface waves, or ripples, that can be detected by a wide variety of non-visual eavesdroppers, such as bats, fish, and even aquatic invertebrates[23–25]. Here we hypothesise that the costs of calling from these risky locations (i.e. floating on water surfaces) is balanced by the benefits obtained through improved signal production. Many frog species produce calls by shuttling a large body of air from their lungs to a vocal sac[26], and frogs may need to float on a water surface to allow for unobstructed inflation of their lungs and vocal sacs.

We address the role of call site characteristics on signal production in túngara frogs (Physalaemus [=Engystomops] pustulosus), a Neotropical species that attracts females with its calls while floating in shallow water bodies that are temporarily formed after heavy rains (water bodies are typically a few centimetres in depth). Males can increase the conspicuousness of their sexual display and hence attractiveness in two ways: by calling louder and by producing more complex calls (adding elements known as 'chucks' to the first part of their call, known as 'whine')[27, 28]. Call production is strongly related to the morphology of the larynx and the size of the vocal sac, as both call amplitude and call complexity seem to depend on the amount and/or speed of airflow (and thus on the maximum volume of air in the lungs)[28]. Prior to calling, males typically sit at their call site, take up air into their lungs, and only start to call when they are floating and their lungs fully inflated. Occasionally, males are observed in the field in very shallow water (<1 cm). These males typically deepen their call sites by rotating and kicking their hind legs before they start inflating their lungs and floating. These observations made us speculate that males need a minimum amount of water to float and take up air into their lungs. Very shallow water could consequently limit floating, the inflation of the lungs, the inflation of this species' large vocal sac, or the shuttling of air between them and thereby influence signal production and attractiveness.

In our present study we assessed variation in water depth of natural call sites and experimentally determined the influence of water depth on call behaviour. We found a positive relationship between water depth treatment, floating behaviour, and the maximum inflation of the lungs and/or vocal sac. Furthermore, we found these changes in calling behaviour to be related to signal amplitude and complexity and consequently male attractiveness. Finally, we used high-speed video recordings and laser-Doppler vibrometry of calling males to demonstrate that maximum vocal sac inflation coincided and resonated with production of complex calls.

## Results

**Observation of calling behaviour in the field.** We found males calling from a wide variety of water bodies, including ditches next to roads, small pools formed on muddy or sandy substrate after heavy rain, as well as large swampy areas. We quantified natural variation in call site characteristics at these locations by measuring water depth at the exact calling position as well as at nearby random positions within the water body. Call sites ranged in depth from 0.2 to 3.7 cm (mean ± s.d. = 1.39 ± 0.65; $n = 126$). Additionally, we found a subset of males had excavated a small puddle ($n = 56$, average diameter 7.56 ± 0.31 cm), which increased the water depth from relatively shallow (average depth of their water body = 0.94 ± 0.38 cm;) to relatively deep (average depth of the manipulated puddle = 1.55 ± 0.44 cm).

**Lowest water depth levels constrain call behaviour.** We collected males from the field and recorded their calling behaviour in our experimental setup, which allowed us full control over the water levels. Male calling was either recorded in response to a rival sound, a large chorus, or no sound and water levels in the setup were randomly altered in between trials.

Forcing males to call at different water depths in our experimental setup had a strong impact on their display behaviour. Floating was strongly affected by water level (Friedmann's test; $n = 20$ males; $\chi^2 = 122.41$; d.f. = 3; $P < 0.001$; Supplementary Fig. 1). At the lowest water levels (0.25–0.5 cm), males often had all legs touching the ground and no male was able to float with all four legs raised. At the highest water levels (1–2 cm) males almost always floated completely (Supplementary Fig. 1). We occasionally observed males rotating and kicking their hind legs in our setup. In the field this kicking behaviour is mainly associated with digging a deeper puddle (Supplementary Movie 1). Water depth also affected the number of hind leg kicks ($\chi^2 = 19.19$; d.f. = 3; $P < 0.001$) and number of rotations ($\chi^2 = 56.89$; d.f. = 3; $P < 0.001$), but not the number of front leg kicks ($\chi^2 = 0.97$; d.f. = 3; $P = 0.81$). In particular in the 0.25 cm treatment, males were continuously rotating and kicking their hind legs.

Water depth treatment also had a strong effect on the inflation of the lungs and the vocal sac (Fig. 1; Supplementary Table 1; Supplementary Movie 2). At the highest water levels (1–2 cm), males showed the largest body width ($\chi^2 = 48.50$; d.f. = 1; $P < 0.001$; a measure that strongly varies with lung inflation) and maximum inflated their vocal sac ($\chi^2 = 62.09$; d.f. = 1; $P < 0.001$). When a male called at the lowest water levels (0.25–0.5 cm), he would occasionally lift off from the ground, propel forward and sometimes hit the wires of the cage with such force that he would bounce back (Supplementary Movie 1). This propelling was not observed for the highest levels of water depth ($\chi^2 = 30.11$; d.f. = 3; $P < 0.001$; Supplementary Fig. 1).

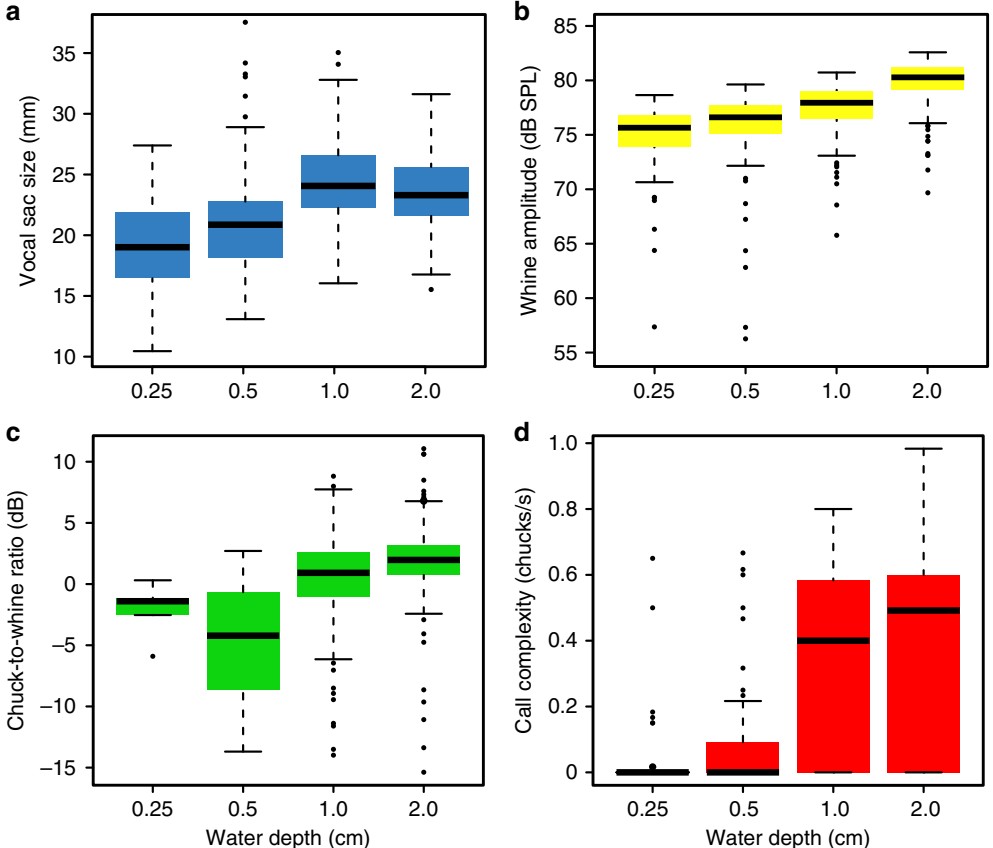

**Fig. 1** Water depth limits signal production. Water depth treatment influenced vocal sac width at maximum inflation **a**. Males inflated their sacs (and lungs) to lesser extent in shallow water. Water depth treatment also affected peak-to-peak amplitude of the whine component of the call (**b**), the chuck-to-whine ratio (**c**), as well as the total number of chucks produced (**d**). Males in deeper water were able to call louder, with more and relatively louder chucks. Boxplots are based on raw data and x-axes depict different water depth treatments. All call parameters increase linearly with water depth (GLMM, all $P < 0.001$, see also Supplementary Table 1)

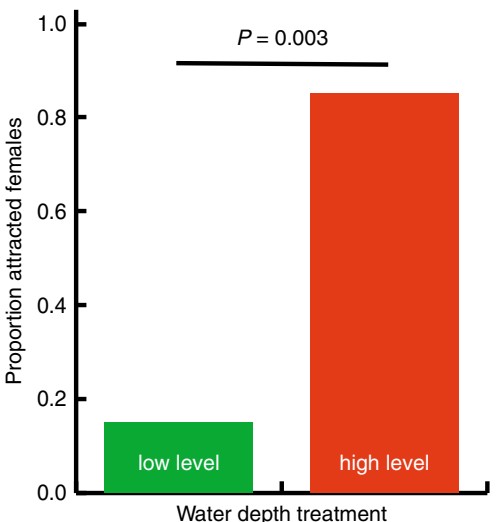

**Fig. 2** Females prefer males that call from the highest water levels. Results from choice experiments with 20 females. The *P*-value was calculated using a binomial test

**Shallow water reduces signal attractiveness.** Males called during 81% of the trials and less at the lowest levels compared to the highest levels of water depth (15–30% of the males did not call at all in water levels of 0.25 and 0.5 cm, whereas all males called at higher water levels). Consequently, call rate was strongly affected by water depth treatment ($\chi^2 = 34.81$; d.f. = 1; $P < 0.001$; Supplementary Table 2). We found the maximum inflation of the lungs to be strongly correlated to maximum inflation of the vocal sac ($b = 1.1$; $R^2 = 0.81$; $P < 0.001$), as well as to the peak amplitude of the frog's whine ($b = 0.83$; $R^2 = 0.26$; $P < 0.001$) and chuck ($b = 0.55$; $R^2 = 0.10$; $P < 0.001$). Water depth treatment consequently had a strong effect on call amplitude and call complexity, with males producing higher amplitude calls ($\chi^2 = 25.85$; d.f. = 1; $P < 0.001$) and making more ($\chi^2 = 28.16$; d.f. = 1; $P < 0.001$) and proportionally higher amplitude chucks ($\chi^2 = 13.75$; d.f. = 1; $P < 0.001$) when calling from higher water levels (Fig. 1; Supplementary Movie 2). Water depth had a stronger effect on the production of chucks when compared to the production of whines. For example, 14 out of 20 males called in very shallow water of 0.25 cm depth, but only 5 out of 20 produced chucks. Contrarily, all males were able to produce complex calls at higher water depth levels (>1 cm). Finally, females showed a strong preference during phonotaxis experiments to approach a focal male when he was calling from deeper water (probability = 0.85; $P = 0.003$; $n = 20$ females, Fig. 2).

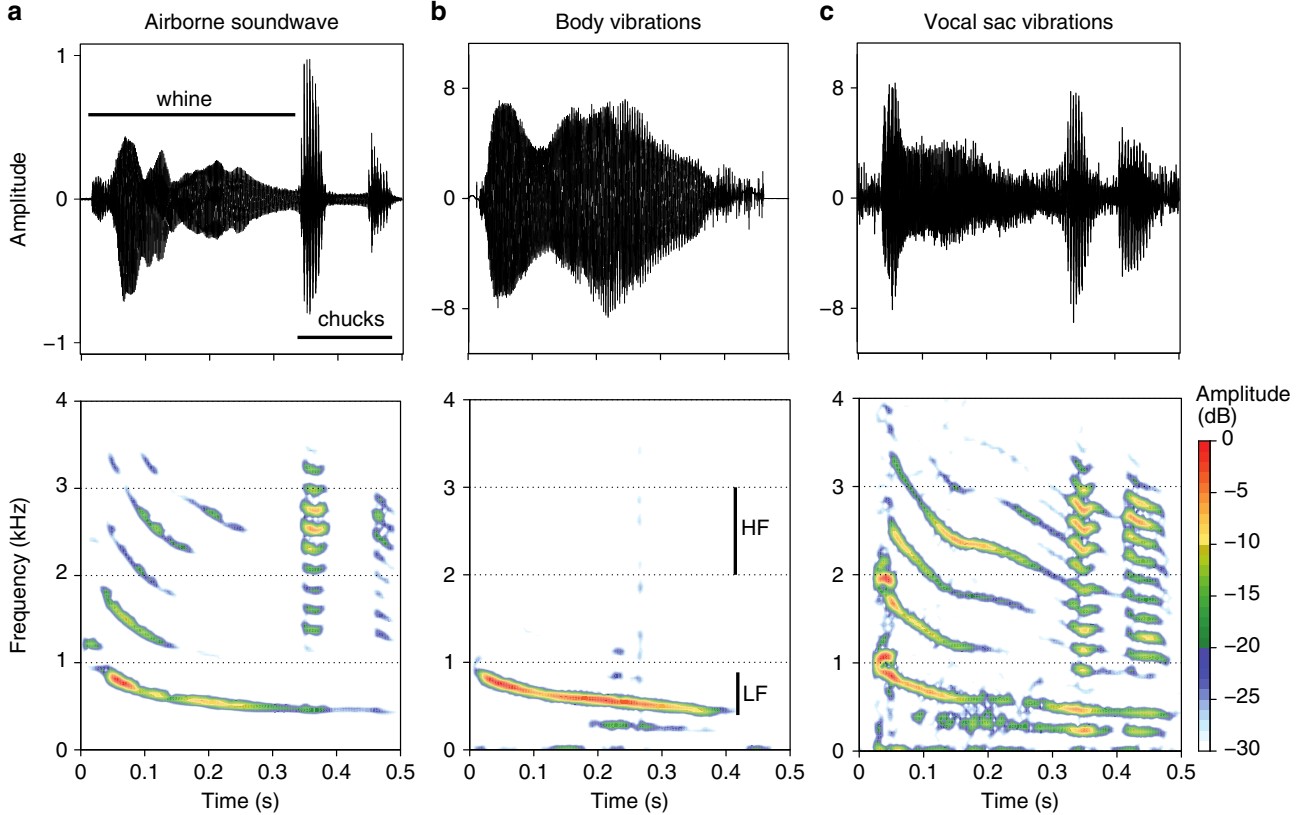

**Fig. 3** The vocal sac is involved in the production of a complex signal. Shown are example recordings of the airborne sound (**a**) made by a male, as well as vibrations of the body wall (**b**) and vocal sac (**c**). Top panels show amplitude envelops of the recorded signals and bottom panels show spectrograms. Both vibrations were recorded shortly after one another from a male that was actively calling with two chucks. Body vibrations were recorded by pointing the laser onto the back of the body, at the position of the lungs. Note that these vibrations lack the higher frequencies (>1 kHz) and lack the chuck component. Vocal sac vibrations were recorded by pointing the laser at the top of the vocal sac, right next to the position of the eye. Note that these vibrations contain substantial energy at the higher frequencies (1–3 kHz) and clearly show the chuck components. Indicated in the mid-panel are the high-frequency (HF) and low-frequency (LF) ranges used to analyse spectral energy of both the whine and chuck vibrations recorded from the frog's body wall and vocal sac (Supplementary Table 2). Y-axes of the top panel depicts normalised sound pressure (**a**), or velocity in cm/s (**b**, **c**)

**Understanding the role of the vocal sac in signal production.** Slow-motion videos of three calling males showed that the maximum vocal sac inflation is reached shortly after the last chuck is produced (Supplementary Movie 3). Vibrations recorded from the vocal sac during calling differed strongly from vibrations recorded from the back of the body (Fig. 3; compare the structure of the whine and the chuck between the vibrations and airborne signal). The vocal sac vibrated with substantial energy at both the high- and low-frequency ranges and vibrated relatively stronger during the production of the chucks when compared to the body wall ($\chi^2 = 15.11$; d.f. = 1; $P < 0.001$; Supplementary Table 2).

## Discussion

Display site properties are an important factor that can influence the production and transmission of mating displays and can thereby drive signal evolution. We show that calling in túngara frogs is strongly influenced by water depth treatment. At the most shallow water depth levels, calling males were unable to float, had less air in their lungs, and inflated their vocal sac to smaller sizes compared to calling in less shallow water levels. Males were continuously kicking their hind legs and rotating in the setup, as if trying to dig a deeper puddle when treated to low water levels. Water depth also influenced the acoustic components of the calling behaviour. Males called at lower rates and produced calls of lower amplitude and complexity in lower water levels

compared to higher levels. Call production thus seems to be influenced by water depth through constraining floating and vocal sac or lung inflation, which in turn limits the production of a loud and complex signal. Such environmental constraints on signal production have important evolutionary consequences, as females strongly preferred a mate that was calling from the higher water levels compared to the lower levels.

Our Laser-Doppler vibrometry measurements of calling males in the field revealed that the vocal sac vibrates with the high-frequency part of both the whine and the chucks, whereas the body wall seems to vibrate mainly with the low-frequency part of the whine. Previous experiments have shown that the spectral characteristics of frogs calling in helium do not change, suggesting that the vocal sac does not act as a resonator[29]. The vocal sac is also much thinner than the body wall[30], which improves the impedance match between the inside and outside air. It thus seems likely that the vocal sac acts as a radiator of the chucks. Additionally, the production of chucks depends on the special morphology of the male's larynx, involving movements of a large fibrous mass that is attached to the vocal folds[28]. Setting this mass into motion may also require high air pressure and/or airflow through the vocal tract and thus a large air volume in the lungs.

Data obtained from lowest water levels (0.25–0.5 cm water depth) suggest that males are constrained by the ability to float and that this constraint in turns affects (1) the amount of air they can take up in their lungs; or, (2) the ability to deal with recoil

forces involved in calling. When males did try to call at the lowest water levels, in particular when stimulated by the playback of chorus sounds, some individuals would occasionally shoot forward, which we interpreted as a movement caused by mechanical forces involved in calling. We presume that, while floating, the water surrounding the frog absorbs the mechanical force, allowing a male to call steadily and from a stable position. Another explanation could be that males need to float when pumping air into their lungs.

Morphological traits have been proposed to constrain signal production and thereby signal evolution[31, 32]. Birds for example need to open and close their beaks to produce song elements with broadband frequencies and the shape of their bill determines how fast they can deliver their individual notes[31, 33]. Birds with heavier beaks are consequently constrained to sing slower songs, or songs covering smaller frequency ranges[33]. Calling in frogs seems to be similarly constrained, as males need to inflate their lungs and/or vocal sac to their maximum volumes in order to produce loud and complex, attractive signals.

Display site properties are known to influence signal design, in particular for species signalling from burrows or tree holes[19–22]. Such correlations may arise when signallers select call sites that show an optimal match between transmission properties and their signal structure. Alternatively, signallers may adjust their signal structure to the transmission properties. Some crickets (*Rufocephalus sp.*), for example, can tune their call frequencies to match the resonance characteristics of their complex structured burrow[34]. Finally, signallers may directly alter transmission properties of their display sites, for example by digging burrows with resonance characteristics matching the dominant frequency of an acoustic signal[19, 22].

Our study demonstrates that environmental conditions can also influence signal production more directly through an interaction with morphological traits. We show that call site properties can set limits to the signal production mechanism, as frogs in our study need to call from relatively deep water in order to float and to inflate their lungs and vocal sacs to the maximum size. Interestingly, these environmental limitations to signal production result in a trade-off for túngara frogs between sexual attraction and predation risk. Bats, among other predators, can use the water ripples induced by calling for detecting males and ripple propagation increases with water depth[23, 24]. Thus, males are forced to choose between signalling from safe, yet unattractive sites, or dangerous attractive ones. This environmentally induced trade-off may result in several evolutionary outcomes. Frogs could either evolve signal components that enhance attractiveness, yet do not depend on morphological structures that can be limited by the environment. Alternatively, frogs may actively alter their display sites, which we occasionally observe in the field. By digging into the mud of shallow water puddles males can increase water depth and thereby their attractiveness, without losing out on increased predation risk.

In conclusion, we show that frogs call from sites with different water depths, which affects the natural as well as sexual selection pressures operating on them. By calling from relatively shallow water, frogs are safer from predators or parasites that can eavesdrop on their call-induced water surface waves. However, we demonstrate that a choice for shallower call sites comes at a large cost of reduced female attractiveness, most likely through production constraints. Irrespective of the underlying mechanism, our results thus demonstrate that call site characteristics can limit the production of a conspicuous signal and can thereby influence signal evolution. Similar processes may also play an important role in other systems where signallers are confronted with confined spaces, such as holes, burrows, branches and twigs. Acoustic signallers, such as crickets[34], toads[19, 20], or parrots[35] can be

constrained by the amplitude at which they can move or inflate their vocal apparatus. Even visual signallers, for example lizards displaying with colourful dewlaps from small twigs may experience limitations in the amplitude with which they can display[36]. By choosing or physically engineering a particular location, signallers are able to alter the balance between sexual and natural selection pressures operating on their displays. Display site properties and the ability to alter them are however likely to differ between habitats. Consequently, rapid habitat alteration, for example in relation to deforestation or urbanisation, may impose novel constraints on signal production and in turn affect selection pressures operating on mating signals.

## Methods

**Study sites and species.** The study was carried out in Soberanía National Park, near Gamboa, Republic of Panama. We focused on male túngara frogs (*Physalaemus* [=*Engystomops*] *pustulosus*), a species that calls from stagnant water bodies formed on the floor of tropical forests after heavy rain. The advertisement call of túngara frogs always starts with an, on average, 400 ms long, harmonic element that decreases in frequency from roughly 900–400 Hz. This element, also known as the 'whine' is often followed by one or more amplitude modulated elements, known as 'chucks', that are typically much shorter in duration (10–20 ms) and of higher dominant frequency (~ 2.3 kHz) than the whine (Fig. 1a). All males can produce these chucks and typically do so when confronted with the calls of a rival male, or the presence of a fertile female.

We collected field data on male calling in May 2016 and June 2017 and we conducted experiments in August 2014 and May 2017 in our lab in Gamboa. For the experiments we collected calling male frogs or pairs in amplexus from various locations in the vicinity of Gamboa 1–3 h after sunset. Frogs were brought to the lab in small plastic containers and were kept in a plastic cooler in between experiments. Males and females were toe-clipped for individual recognition after the experiment and their mass and snout-vent length (SVL) were measured. All individuals were released back to site from which they were collected on the same night. All experiments with frogs were licensed and approved by STRI (IACUC permit: 2014-0805-2017) and the Autoridad Nacional del Ambiente de Panama (SE/A-82–14).

**Experimental procedures.** Male frogs were tested individually in a hemi-anechoic chamber under IR-lighting. At the start of the experiment they were placed in a small cage consisting of a ring of evenly spaced nylon monofilament (diameter of 0.05 mm fishing line every 0.5 cm). The cage consisted of a circular plastic base (diameter of 8 cm) and a circular plastic top ring (diameter of 8 cm, with a ⌀ 4 cm hole in the centre) that was supported by three metal rods (diameter of 0.4 cm). The cage was placed in a pool (diameter of 50 cm) containing a tube that allowed the experimenter to either add (using a funnel) or subtract (using a 50 ml syringe) water, in order to manipulate water depth at the position of the frog.

Males were stimulated with a low-amplitude chorus recording until they were readily calling for 1 min. Males that did not start to call within 10 min were excluded from the experiment ($n = 5$; 20% of total males tested). We assessed male calling behaviour at 4 different water depths (0.25, 0.5, 1.0, and 2.0 cm) during three different acoustic treatments (chorus sounds, single call, and silence). All sounds were broadcast from a Peerless full-range speaker (Tympany, TG9FD10-08, 3.5 inch, 8 ohm, Sausalito, CA, USA) connected to a desktop PC and a NAD 40 × 2 W amplifier (NAD electronics, Ontario, Canada). The speaker was placed 20 cm behind the focal frog and 65 cm from the microphone. For the single call we used a synthetic signal consisting of a whine plus one chuck broadcast at 0.5 calls/s (see ref. [37] for details of signal construction). All sounds were broadcast at 82 dB SPL (re. 20 μPa at 50 cm, measured with Extech Instruments SPL-metre set to C-weighted, fast and max, type 407764, Nashua, NH, USA). All males were tested on a full-factorial randomised and balanced design, which resulted in a total of 12 trials. Each trial lasted for 1 min, followed by a 2-min break during which the water depth was altered (starting 1 min before the next trial). All males received the 12 trials in a row.

We assessed whether water depth influenced a male's attractiveness to females in a hemi-anechoic chamber under IR-lighting. Individual females were placed under a funnel in the centre of the room (2.7 × 1.8 × 1.78 m, L × W × H; Acoustic Systems, Austin, TX, USA) and were stimulated with a playback from two opposing speakers (80 cm away from the female). These speakers played the call from one male recorded in either low or high water levels antiphonally with a call rate of 1 call/2 s. For different stimuli we selected recordings with a minimum of 6 calls in 60 s of a male calling in the low level (not all males called in the 0.25 or 0.5 cm treatment, we therefore selected the recording with highest call rate from the two lowest water level treatments) and high level treatment (all taken from the 2.0 cm treatment) for 10 different males. From these recordings we selected one call that was of maximum complexity (highest number of chucks). Acoustic features of the playback stimuli were not manipulated to test for their combined effects on female preference. The speakers were calibrated with a reference recording of an artificial

whine set to 90 dB SPL (re. 20 µPa at 50 cm, measured with the Extech SPL-metre set to C-weighting, fast and max). This reference signal had an amplitude ~ 6 dB greater than the highest amplitude call used in the preference test (we used natural variation in amplitude). We raised the funnel after 2 min from the start of the playback and scored the female's choice (defined as staying 2 s within a 10 cm radius of the centre of a speaker). When a female did not move during the first 5 min of the test or any subsequent 2 min we ceased the experiment and did not retest this female ($n = 6$). All females that did choose made a clear choice for one of the two speakers within 10 min.

**Data collection**. We collected data on the water depth at natural calls sites from 5 different locations by sticking a ruler vertically at the position of a calling male. We sometimes observed males around dusk at our field sites that would first deepen their call site by rotating and kicking of their hind legs before they would start to call. This digging behaviour resulted in a characteristic circular depression in the mud. For these males we also measured the water depth at the edge of these puddles in a randomly determined direction (thus at a location not modified by the frog).

In the lab we recorded male movements with a camera that was mounted on top of the cage (mini 1/4″ CCTV camera; 2.8 mm lens; connected to a desktop PC). We recorded male calls with a microphone setup (G.R.A.S. 40 BF microphone amplified by 20 dB by G.R.A.S. 26 AC amplifier connected to an Avisoft 116Hm Ultrasound gate, G.R.A.S. Sound & Vibration A/S, Holte, Denmark) onto a desktop PC, using a sampling rate of 50 kHz. The microphone was placed at a 45° angle and at a distance of 50 cm from the frog. The microphone was calibrated prior to each experiment using a tone generator (G.R.A.S. 42 AB, 114 dB at 1 kHz).

In addition to the main experiment we recorded the calling behaviour of several males in more detail to obtain a better understanding of mechanisms underlying signal production and in particular to assess the role of the vocal sac inflation in the production of chucks. We filmed 3 male frogs in our lab setup using a high-speed video camera (Optronis GmbH, 500 fps; CamRecord CR600 × 2, Kehl, Germany) from the front as well as from the side. Additionally we recorded the body vibrations of 6 males that were calling in the field using a laser-Doppler vibrometer (PDV-100, Polytec, Waldbron, Germany, set to 20 mm/s/V, LDV here after) connected to a field-recorder (Marantz PMD660, 44.1 kHz sampling rate, 16 bit sampling depth). Recording levels of the LDV were calibrated before and after each recording using a build-in 4 V reference signal. The LDV was mounted on a tripod that was placed over a calling frog and with the laser beam either pointed to the back of the frog, midway between its head and vent, or to its vocal sac at either side of the head. The position of the laser was slightly adjusted until we reached satisfactory signal-to-noise levels. In addition to recording from above we also recorded two individuals from the side, to ensure that we were recording body vibrations and not vibrations of the water surface underneath the frog.

**Data analyses**. We analysed the videos of male calling in our setup and scored whether males were completely floating (score = 2), partially floating (hind legs touching the bottom of the setup; score = 1) or not floating at all (score = 0). Additionally, we noted that males would sometimes try to dig a small puddle for themselves, a behaviour that we often observe in the field during which males kick their hind legs into the mud while rotating. We therefore scored the number of hind and front leg kicks and rotations per trial. We also observed that males would occasionally shoot forward when calling in shallow water. In some occasions males would even hit the nylon threads of their cage and bounce back, suggesting they put substantial force in their call effort. We therefore also scored the number of times males propelled themselves per trial.

We selected for each trial three video stills from the beginning, middle and end of a call bout. We selected for each call the video still with the maximum inflation of the lungs as well as maximum inflation of the vocal sac. We used the programme ImageJ[38] to measure from these images the maximum body width at the position of the lungs (when resting in between calls) as well as the width of the vocal sac at maximum inflation. Additionally we measured the snout-vent length in ImageJ to calibrate (in mm) both morphological measurements with our previous measurements of the snout-vent length measured on the animal prior to experiments.

Sound and body vibrations were analysed in SASLab Pro (Avisoft Bioacoustics, Berlin, Germany). For the calls recorded at different water depths we counted the total number of whines as well as the total number of chucks made during each of the 1-min trials. Additionally we selected three calls from the start, middle and end of a trial and measured the peak-to-peak amplitude of the whine and chuck part of the call separately and used these values to calculate the chuck-to-whine ratio.

For analysing body vibrations we selected up to three calls from each male with sufficient signal-to-noise ratio. We divided each call in two parts of equal length and measured RMS (root-mean-square) amplitude in the first (overlapping mostly with the whine) and the second part (overlapping mostly with the chucks) using the programme Matlab (the Mathworks, 2015). Additionally we filtered calls in a low-frequency (0.4–0.9 kHz; whine) and high-frequency range (2–3 kHz; chuck) and measured spectral energy (RMS amplitude) in these different ranges. All amplitude values were used to calculate the ratio in spectral energy between the whine and chuck part of the call and were averaged per male.

We assessed the effect of water depth and sound treatment on male calling behaviour in $R$ (v.3.2.2)[39]. Linear mixed models were constructed using the package lme4. We modelled random variation in the effect of explanatory parameters per individual frog by fitting random slopes as well as intercepts and selected the best random structure using AIC scores[40]. Furthermore, we modelled trial number by fitting random intercepts. The models on number of whines (expressed as elements/s), number of chucks (elements/s), vocal sac width, body width, peak amplitude and chuck-to-whine ratio all contained water depth and sound treatment as fixed effect and used a Gaussian distribution with identity link function. The interaction between water depth and sound treatment was not significant for any of the call parameters and was therefore left out of the final models. Models on vocal sac and body width, as well as amplitude and chuck-to-whine ratio contained call order (start, mid or end section of a trial) as a random intercept. The amplitude ratio in body vibrations used a Gaussian distribution with log link-function and contained recording location (body wall or vocal sac) as fixed effect. Significance of fixed effect was assessed using likelihood (ML) ratio test. All models were tested for normality, overdispersion and heteroscedasticy. Data on floating, kicking and propelling behaviour did not fit model assumptions and were analysed with a non-parametric Friedmann's test. Female preference was analysed with a binomial test.

**Data availability**. Raw data on male call behaviour, female preference test and call site characteristics can be found at the Dryad online depository (DOI:10.5061/dryad.kc4p4).

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

## Acknowledgements

We are grateful to Dylan Gomes for his help on the vocal sac measurements and to Meghan Still for her advise on the female preference experiment. We thank Marco Tschapka and the University of Ulm, Germany for letting us use the high-speed camera. We also thank the Smithsonian Tropical Research Institute (STRI) for use of their Laser vibrometer. Comments from Sandra Goutte substantially improved the manuscript. The research was funded through a Smithsonian fellowship and a Marie Curie grant (#655262) to W.H.

## Author contributions

W.H. conceived the study, designed the experiments, made sound and laser recordings, analysed the data and wrote the paper. H.L. and J.A.H.S. collected field data on call sites and carried out female choice experiments. A.M.L. advised on experiment and discussed the results. I.G. recorded the high-speed videos and discussed the results. J.E. wrote the paper. M.J.R. wrote the paper.

## Additional information

**Competing interests:** The authors declare no competing financial interests.

