## [Peer Review File · Nature Communications]

Reviewers' comments:

Reviewer #1 (Remarks to the Author):

This is a fascinating study of how animals signals can be influenced by the environment and, given this, how animals might alter their signaling behaviors to maximize their attractiveness. The study is well written and was fun to read. The authors make a step-by-step case from field observations that frogs in shallower pools often dig holes to increase depth, to experiments showing that depth affects attractiveness of calls in multiple ways, to female preference test showing that females prefer calls from deeper pools. There is an inference that calls from males who dig deeper pools in the field will be more attractive than calls from males in pools of similar initial (pre-digging) depths who do not, or that digging in natural pools increases the attractiveness of subsequent calls of the same male. But this inference is well supported by the data. The methods are rigorous and use the appropriate technology for addressing these questions with high precision. As the authors clearly acknowledge, this is not the first study to demonstrate that the environment can affect signals, or that males adjust their signaling behavior, or that resulting signals are more attractive. But it is among the first to pull these all together in one study system, and to include an experimental and field component. A number of recent review papers have addressed issues of signal complexity, plasticity in display behaviors in response to the environment and social context, and how these factors might affect signal evolution. Therefore the study is timely.

My other comments are minor.

Line 100-103: This section needs clarification. How can males who have altered their sites be at 9.4 cm, as this seems to be outside the range of depths mentioned previously? Is this a typo? What is the 1.55 cm number of line 103 referring to? The difference between those who dug and those who did not, or between the deepest part that was modified by the frog versus the natural depth? (I assume the latter, based on the Methods section, but this should be clearer here.) What is the depth of pools for frogs who did not dig? A logistic regression here would help to make the case that the probability of digging decreases with pool depth.

Line 141: Include the numbers for deep water for comparison

Line 232: Change loosing to losing

Line 251-258: This comes a little out of left field. It's mentioned so briefly that it's unclear what point you're trying to make. Are you arguing that this is a conservation concern which your results highlight, or simply noting that signals may be different in cities and other areas where the habitat cannot be modified. Are these frogs common in urban areas? Are they of conservation concern? Are chucks less common in cities? Is the digging behavior common enough among species that this might be a factor in determining we can adjust to urbanization? The same can be asked about studies of bird songs in urban areas, so I'm not questioning whether this possibility is of interest. But I think this point needs to be clarified. I suggest shortening it rather than expanding it, to avoid ending on such a speculative note.

Methods:

Line 306: A little more detail is needed. How long between trials? Was there a fixed time? Did you run all 12 trials in a row on each individual?

Reviewer #2 (Remarks to the Author):

I generally like this study showing the influence of calling site characteristics on calling behaviour and call characteristics. However I have two main reservations against the manuscript in its current form that the authors would have to address in a revision (or in a profound rebuttal):

1) In my opinion, the entire focus of the argument about "water depth" as the crucial influence on calling gives the wrong impression of a direct (linear) relationship/correlation between water depth and the parameters measured. However, during reading it becomes clear, that this is more than anything an indirect relationship – to me the actual influence on calling seems to be the difference between "floating" vs. "non-floating" which only then is a function of water depth. I already made a corresponding remark after at L86, without knowing your experimental setup. Then I was assuming, also based on specific wording in the introduction, that you were investigating the influence of water depth on floating frogs – in actually **really** deep water, compared to your actual setup. Only later it became clear, that your experimental setup (and thus your hypotheses) is placed at the transition from non-floating to floating, in, comparatively, very shallow water! From here, the term "deep" water is highly confusing and misleading throughout the manuscript, as it suggests conditions totally different than they actually were in your experiment and for frogs in the wild, that were sitting in a few centimetres of water. Also your display of water depth on a logarithmic scale and with a linear regression in the graphs is prone to obscuring the actual nature of the effect, which to me rather is a phase-transition/tipping point from non-floating to floating.

I believe that a shift of the entire perspective on calling/floating frogs on the one aspect that really matters (in my opinion) – floating vs. non-floating – would tremendously help in outlining and emphasizing the underlying physical principles and constraints of/on frog calling and give an image truer to the actual effects acting. Of course you still would have to add that floating vs. non-floating is a (rather non-linear) function of water depth, but this is rather trivial and directly derived from simple physical laws. ...maybe there is a trade-off with body size and mass...

2) As you can see from the by-line comments, the manuscript, especially in the materials and methods, but also in the introduction and discussion, suffers over wide stretches from omissions of relevant details, and from assumptions/information that are not evident for a reader that is not familiar with the study species. For the discussion, this is partially also the case because of the journal's unfortunate style to put the methods at the end of the manuscript, which actively hides relevant information from the reader, who then has to jump back and forth to be able to understand what was actually done in the study. To overcome this imposed handicap, please try to identify such relevant information from the materials and methods and present it in the introduction and discussion where adequate, even at the risk of being redundant.

It would also be nice to get a little more background and natural history information on your study species – in the dedicated species description, as well as interwoven throughout the manuscript. Feel free to cite excessively from the comprehensive work of your group...after all, you are studying an organism proper, not a "system"...

L31: "...which can play **a** crucial role..."

L46: "...from **where** they display."

L46: Please phrase - "To change one's signalling location" is conceptually not comparable to the attempts at increasing attractiveness presented in the previous sentence. As laid out in the preceding sentence, attractiveness is not improved "by changing location" as such, but by accompanying

improvements in visibility, signal transmission, contrast, background noise...etc. So changing location serves specific purposes, but is not the purpose itself, as the wording suggests, and thus no "alternative" to the adjustments in signalling behaviour.

L49: "...displaying *in* different water depths..." – "from" suggests that displaying is directed "to" somewhere else – like out of the water or to the water's surface, whereas here the signals receiver is, most widely, "in" the same water depths as the signaller"

L74: Please elaborate a little on how "frogs floating on a water surface may be facilitating the mechanical forces associated with this behaviour." or at least give a reference to give an idea what you actually mean by that.

L79: I might be wrong, but I believe outdated taxonomic names are rather presented like Engystomops (= Physalaemus) pustulosus

L79: Awkward wording, please rephrase "...calls from puddles during floating" or something alike.

L86ff: Cf. comment on L74 – please elaborate on the physical rationale behind these hypotheses (or clarify the assumptions); to me they are not self-explanatory nor do they follow stringently from physical conditions. If the frog is already *afloat* on the water surface (and not just "in" the water, but actually touching the ground) before calling (as suggested by L79), any expansion of the vocal sac would increase its volume/decrease its average density and should actually lift the frog(its centre of gravity)/the vocal sac *out* of the water, which should happen independently of water depth! Only in very shallow water, with the frog sitting on the ground, expansion during calling would actually lead to increased ground contact and the expansion having to act against gravity on the frog's mass.

After having read the entire paper (and especially the methods hidden at the end...), all this becomes clearer – you actually rather tested the effects of floating vs. not floating, and from which water depth on frogs will float. Please rephrase all relevant passages accordingly so this becomes clearer, that you did not assume/test a linear/graded influence of water depth on calling of *floating* frogs (in water that was sufficiently deep for floating), but rather influences at the transition from floating to non-floating.

L98: I pity the authors (and readers) for having to deal with this journals cumbersome style of banishing the methods to the end of the paper. This is a real curse for the flow of reading and understanding ones work!

L100ff: I don't fully understand how these numbers are supposed to be read and how they add up. Relative to what were the water bodies of males that modified their call site "shallow" – as far as I understood they actually *deepened* their site (9.4 vs 1.39 cm)?! What was 1.55 +/- 0.44 cm – was this the average difference between natural and modified puddles, or the increase in depth frogs in the very puddles the frogs deepened?

L103: please rephrase – the frogs did not create deeper puddles as a consequence *by* calling form shallow water bodies, as your wording suggests sensu stricto.

L107: Please reconsider and rephrase accordingly - as following from simple physical laws, floating is rather a strict function of water depth, and size and buoyancy of the frogs, I don't understand why this is considered an "ability of males", nor why this even requires testing. The actually interesting information is rather, which water depth is required to ensure floating for the average Tungara, and the passage should be rephrased and presented accordingly.

L109ff: And how about deep water? I miss the corresponding information!?

L159: In my opinion the focus on "water depth", also in the discussion, gives the entire paper a twist in the wrong direction. The actual effect on calling activity and characteristics comes from floating vs. not-floating which only then is (mostly) a function of water depth (and to a lesser extent to animal size and density).

L251: Change "our species" against the scientific or vernacular name

L252: Isn't this only the case for one of the signals modalities (water ripples), whereas in the other modality (airborne sound) actually the conspicuousness is (still) increased – as it is intended to increase the attractiveness for females?! Cf. L55 where you say that "signal components" can be used by predators for detection. Or are tungaras safe from airborne acoustic eavesdropping?

L259: Can you elaborate a little on the specifics of "concrete structures associated with the drainage system", especially the water levels tungaras can expect there?

L264: Cf. L79 – at least please be consistent with your nomenclature!

L270, check throughout: "10–20" – use – (dash) not – hyphen to indicate ranges

L279: Is toe-clipping strictly necessary for individual recognition of Tungara frogs or would there have been other feasible methods like pattern matching? Was individual identification for this study the only purpose of this toe clipping, or were tissue samples collected also for other purposes? Was toe-clipping only for marking specifically mentioned/allowed in your experimental permits? Can you present evidence of the effects of toe-clipping in this species? Please explain.

L298: I assume water depths of 0.25 – 2.0 cm ?!+

L300: Please specify the *acoustic* characteristics of the speaker and also give information on the playback device; here and elsewhere, give complete maker information (type, company, city, (state), country).

L301: does "synthetic" refer to a signal that was digitally created from scratch, or one that was composed from recorded parts. Please specify the characteristics of the signal in more detail and how/why they were chosen (e.g. population average, etc.)? Please also give specifications and origin of the chorus stimulus.

L310: please specify the dimensions of the room and how far the speakers were away from the tested female.

L311ff: Your wording is confusing to understand the actual experimental setup and its rationale – you recorded male frogs at 4 water depths, but used only recordings from 3 depths in the playbacks (why?) with females, but categorize them as only shallow (2 different depths) or deep (only 1 depth) (why?). This is confusing and needs clearer writing.

L312: Your wording "minimum of 6 calls" suggests that you used stimuli with a varying number of calls per time? What was the actual range/average of the call rate? Why did you not manipulate the stimuli to standardize the call rate across stimulus conditions? In the introduction you did not explicitly suggest call rate as a potential function of water depth – did you actually see/treat call rate as a part

of the experimental condition, or could it be an overlooked confounding effect on female preference? If you don't think so, please lay out your rationale.

After having read the entire paper it becomes clear, that you actually do see also call rate as a function of floating vs. non-floating (water depth) – this should be made clear much earlier in the manuscript

L318: There is missing a lot of information here: How long was the overall trial length? For how long was the acoustic stimulus presented – or did you simply directly use the 1 min recordings from the male trials? If you just used the 1 min recording, does this mean there was 1 min of silence before you lifted the funnel – or did you play the 1 min recording twice? Did you continue playing the stimulus after the funnel was lifted, or did you stop the playback and the females were still deciding on one side although there was no stimulus anymore? Did you stop the trials after a defined time when females did not move? If so, how many did not move? Did all females immediately and unambiguously decide for a speaker or did you have a criterion in place to score decisions?

L321: I suggest the heading "Data collection in the field"

L326: the diameter of the dug-out puddle is rather a result, no?

L327: Unclear reflexive wording – does "these puddles" here refer to the larger, natural body of water, or to the excavation by the frogs?

Unclear wording - Naturally, the "depth at the edge of a puddle" converges against 0, unless you have a straight wall perpendicular to the water surface. Also I don't understand, how you can "measure depth...in a predetermined direction" – do you want to say that you measured the depth of (the natural) puddles at a defined (how much?) distance from the edge, in a defined (which?) direction? Please rephrase for clarity.

L330: I suggest to integrate the specification of the technical equipment in the experimental setup into the according paragraphs (L285ff)

L331ff: complete maker information please

L334: "...at an angle of 45° to the water surface, and at a distance..."

L343: There it is - this is complete maker information for the first time!

L347: "16 bit sampling depth"

L349-350: delete one "either"

L356ff: Rephrase for clarity – "the videos" - as it reads now, it only becomes clear at the end of the paragraph, when you mention the cage, that this is about filming of calling males in the laboratory, and not about the high speed filming under natural conditions.

L370: Please give a proper reference for ImageJ as per the suggestion of its authors:
<https://imagej.nih.gov/ij/docs/faqs.html>

L383: In the context of this study, I do not understand where and why normalization was needed – please explain. What do you mean by "for visual purposes"? I can only assume that this refers to L483 – the envelopes and spectrograms of normalized calls and vibrations? If this is the case, leave it out

here – an explanation in the figures legend is sufficient.

L386/397: "...up to three calls from each male..." suggests, that you averaged amplitude measurements within males, or that "individual male" was a random factor in your model, but this is not clearly stated.

L390-391: cf. L270, please be consistent with the format of reporting ranges

L396: Please give a proper reference for R as per the authors suggestion: <https://cran.r-project.org/doc/FAQ/R-FAQ.html#Citing-R>

L457: The somewhat hidden logarithmic x-axis (doubling of the distance with every tick) is misleading in suggesting a strong and linear relationship between water depth and call parameters; actually the effect is far less than suggested and it is not linear but rather logarithmic/asymptotic. I rather see the characteristics of a phase-transition/tipping point from non-floating to floating (which then depends on water depth) than a direct influence of water depth on the call parameters. This is especially the case for "call complexity". Returning to L395 - how was water depth coded for the linear mixed model – on an ordinal or on a linear scale?

Reply to reviewers comments:

Reviewer #1 (Remarks to the Author):

This is a fascinating study of how animals signals can be influenced by the environment and, given this, how animals might alter their signaling behaviors to maximize their attractiveness. The study is well written and was fun to read. The authors make a step-by-step case from field observations that frogs in shallower pools often dig holes to increase depth, to experiments showing that depth affects attractiveness of calls in multiple ways, to female preference test showing that females prefer calls from deeper pools. There is an inference that calls from males who dig deeper pools in the field will be more attractive than calls from males in pools of similar initial (pre-digging) depths who do not, or that digging in natural pools increases the attractiveness of subsequent calls of the same male. But this inference is well supported by the data. The methods are rigorous and use the appropriate technology for addressing these questions with high precision. As the authors clearly acknowledge, this is not the first study to demonstrate that the environment can affect signals, or that males adjust their signaling behavior, or that resulting signals are more attractive. But it is among the first to pull these all together in one study system, and to include an experimental and field component. A number of recent review papers have addressed issues of signal complexity, plasticity in display behaviors in response to the environment and social context, and how these factors might affect signal evolution. Therefore the study is timely.

We thank the reviewer for his/her kind words and clarification why our study is timely. We agree that we do not claim to be the first to reveal an environmental impact on sexual signal design. However, we do argue that the majority of previous studies focused on effect of transmission or perception on signal evolution, rather than a constraint on production.

My other comments are minor.

Line 100-103: This section needs clarification. How can males who have altered their sites be at 9.4 cm, as this seems to be outside the range of depths mentioned previously? Is this a typo? What is the 1.55 cm number of line 103 referring to? The difference between those who dug and those who did not, or between the deepest part that was modified by the frog versus the natural depth? (I assume the latter, based on the Methods section, but this should be clearer here.) What is the depth of pools for frogs who did not dig? A logistic regression here would help to make the case that the probability of digging decreases with pool depth.

We have clarified this section and thank the reviewer for catching the typo.

Line 141: Include the numbers for deep water for comparison

We had added results for deep water

Line 232: Change loosing to losing

We have changed it, thanks for catching!

Line 251-258: This comes a little out of left field. It's mentioned so briefly that it's unclear what point you're trying to make. Are you arguing that this is a conservation concern which your results highlight, or simply noting that signals may be different in cities and other areas where the habitat cannot be modified. Are these frogs common in urban areas? Are they of conservation concern? Are chucks less common in cities? Is the digging behavior common enough among species that this might be a factor in determining who can adjust to urbanization? The same can be asked about studies of bird songs in urban areas, so I'm not questioning whether this possibility is of interest. But I think this point needs to be clarified. I

suggest shortening it rather than expanding it, to avoid ending on such a speculative note.

Our main point in this section is that the environment can impose constraints on signaling through mechanical limitations, which could drive signal evolution. The substrate from which animals display in cities is often very different from the substrate in rural areas, which in some cases (for example our frogs, or potentially for birds) may affect signaling behavior. We have shortened this section and left out the details with respect to our study species.

Methods:

Line 306: A little more detail is needed. How long between trials? Was there a fixed time? Did you run all 12 trials in a row on each individual?

We have added more detail on the procedure.

Reviewer #2 (Remarks to the Author):

I generally like this study showing the influence of calling site characteristics on calling behaviour and call characteristics. However I have two main reservations against the manuscript in its current form that the authors would have to address in a revision (or in a profound rebuttal):

1) In my opinion, the entire focus of the argument about “water depth” as the crucial influence on calling gives the wrong impression of a direct (linear) relationship/correlation between water depth and the parameters measured. However, during reading it becomes clear, that this is more than anything an indirect relationship – to me the actual influence on calling seems to be the difference between “floating” vs. “non-floating” which only then is a function of water depth. I already made a corresponding remark after at L86, without knowing your experimental setup. Then I was assuming, also based on specific wording in the introduction, that you were investigating the influence of water depth on floating frogs – in actually *really* deep water, compared to your actual setup. Only later it become clear, that your experimental setup (and thus your hypotheses) is placed at the transition from non-floating to floating, in, comparatively, very shallow water! From here, the term “deep” water is highly confusing and misleading throughout the manuscript, as it suggests conditions totally different than they actually were in your experiment and for frogs in the wild, that were sitting in a few centimetres of water. Also your display of water depth on a logarithmic scale and with a linear regression in the graphs is prone to obscuring the actual nature of the effect, which to me rather is a phase-transition/tipping point from non-floating to floating.

We agree with the reviewer that the use of deep vs shallow can be misleading and we apologize for any confusion this may have caused while reading our manuscript. It is clear that most readers would not associate the term 'deep' with water bodies of just a few centimeters. We have rewritten our manuscript to clarify that we use shallower versus deeper in a relative way and we have stated early on that we are focusing on water depths of just a few cm. Where possible, we refer to lower, or lowest levels of water depth versus higher, or highest levels of water depth.

We also appreciated the reviewer's suggestion that floating vs non-floating determines the experimental effect on our call parameters. We have rewritten our introduction and discussion to acknowledge this point. However, we like to point out that our manipulation involved altering water depth, not floating behaviour. Consequently, we can only make conclusions with respect to water depth treatment, even though its effect may act via floating behaviour. We do however realise that it can be misleading when thinking of the physical law(s) underlying the link between water depth and call parameters. We agree that the effect of

water depth operates mainly at the range where the capacity to float is affected. We have rephrased our manuscript in such a way that it is clear what we did (altering water depth) and that it may have an indirect effect on call parameters through floating (effect on floating and amount of air taken up in lungs/vocal sac). We thank the reviewer for his/her suggestions, which made us think carefully about our design and results again, which improved our understanding of the mechanisms that are at play in our study.

I believe that a shift of the entire perspective on calling/floating frogs on the one aspect that really matters (in my opinion) – floating vs. non-floating – would tremendously help in outlining and emphasizing the underlying physical principles and constraints of/on frog calling and give an image truer to the actual effects acting. Of course you still would have to add that floating vs. non-floating is a (rather non-linear) function of water depth, but this is rather trivial and directly derived from simple physical laws. ...maybe there is a trade-off with body size and mass...

See our comments above. In addition, we would like to add that we do not completely understand the mechanism responsible for our observed relation between water depth treatment and call behaviour, as we also describe in our discussion. Furthermore, we believe that the non-linearity in floating behaviour is most likely caused by the categorical way we scored it. We scored a 0 when all legs touched the ground, a 1 when either hind or front legs touched the ground and a 2 when no legs touched the ground. We like to argue that scoring of vocal sac and lung volume provides a much more accurate (and linear) estimate of the effect of water depth on the production mechanism. Also, we are not sure whether floating is directly derived from simple physical laws. It is probably the result of the amount of air taken up by the frog in relation to its body size and as such it is also affected by the animal's behaviour, thereby increasing the complexity in response through the interaction with water depth. Given that we do not fully understand the underlying mechanism we prefer to use water treatment when writing about our results, but we have now put water depth more in relation to floating and the ability to inflate the lungs and vocal sac in the abstract, introduction and discussion. Irrespective of the underlying mechanism, our data show that call site properties can constrain signaling, which is to us the most important take-home message of our paper.

2) As you can see from the by-line comments, the manuscript, especially in the materials and methods, but also in the introduction and discussion, suffers over wide stretches from omissions of relevant details, and from assumptions/information that are not evident for a reader that is not familiar with the study species. For the discussion, this is partially also the case because of the journals unfortunate style to put the methods at the end of the manuscript, which actively hides relevant information from the reader, who then has to jump back and forth to be able to understand what was actually done in the study. To overcome this imposed handicap, please try to identify such relevant information from the materials and methods and present it in the introduction and discussion where adequate, even at the risk of being redundant.

It would also be nice to get a little more background and natural history information on your study species – in the dedicated species description, as well as interwoven throughout the manuscript. Feel free to cite excessively from the comprehensive work of your group...after all, you are studying an organism proper, not a “system”...

We thank the reviewer for his/her suggestions to improve the readability of our manuscript and we apologize for any inconveniences these omissions may have caused reading the previous version. We have tried to provide sufficient methodological/experimental information throughout the manuscript to better interpret the results and to follow the discussion.

L31: "...which can play *a* crucial role..."

Thanks for catching!

L46: "...from *where* they display."

We changed this

L46: Please phrase - "To change ones signalling location" is conceptually not comparable to the attempts at increasing attractiveness presented in the previous sentence. As laid out in the preceding sentence, attractiveness is not improved "by changing location" as such, but by accompanying improvements in visibility, signal transmission, contrast, background noise...etc. So changing location serves specific purposes, but is not the purpose itself, as the wording suggests, and thus no "alternative" to the adjustments in signalling behaviour.

The conceptual comparison depends on the definition of attractiveness. We refer to attractiveness in its most simple form, which is the ability of a sender to attract a receiver. This ability can be improved by changing the signal, or by changing the location. Both actions could have an effect on the transmission of the signal, or the reception, thereby increase detection and consequently the ability to attract a mate (attractiveness). We have added a phrase to make clear that we also view increased signal efficacy as a way to improve attractiveness.

L49: "...displaying *in* different water depths..." - "from" suggests that displaying is directed "to" somewhere else - like out of the water or to the water's surface, whereas here the signals receiver is, most widely, "in" the same water depths as the signaller"

Thanks for suggesting this improvement.

L74: Please elaborate a little on how "frogs floating on a water surface may be facilitating the mechanical forces associated with this behaviour." or at least give a reference to give an idea what you actually mean by that.

As we have outlined in our reply to the general comment above we do not fully understand the mechanics driving the causal relationship between water depth and call behaviour. We have rephrased this part to make clear that we speculated about the possible impact of water depth on call behaviour based on field observations, not a priori knowledge of physics.

L79: I might be wrong, but I believe outdated taxonomic names are rather presented like Engystomops (= Physalaemus) pustulosus

The most recent review of the phylogenetics of advanced frogs published this year uses physalaemus and not engystomops. It is subject of ongoing debate.

L79: Awkward wording, please rephrase "...calls from puddles during floating" or something alike.

We have rephrased it and provide a first description of the typical water depth at which frogs call.

L86ff: Cf. comment on L74 - please elaborate on the physical rationale behind these hypotheses (or clarify the assumptions); to me they are not self-explanatory nor do they follow stringently from physical conditions. If the frog is already *afloat* on the water surface (and not just "in" the water, but actually touching the ground) before calling (as

suggested by L79), any expansion of the vocal sac would increase its volume/decrease its average density and should actually lift the frog(its centre of gravity)/the vocal sac *out* of the water, which should happen independently of water depth! Only in very shallow water, with the frog sitting on the ground, expansion during calling would actually lead to increased ground contact and the expansion having to act against gravity on the frog's mass.

After having read the entire paper (and especially the methods hidden at the end...), all this becomes clearer – you actually rather tested the effects of floating vs. not floating, and from which water depth on frogs will float. Please rephrase all relevant passages accordingly so this becomes clearer, that you did not assume/test a linear/graded influence of water depth on calling of *floating* frogs (in water that was sufficiently deep for floating), but rather influences at the transition from floating to non-floating.

We now realize what causes the confusion and are grateful for the reviewer for pointing this out. Yes, we believe water depth to be only of influence on signal production within the range where the frog's body is still touching the ground prior or during calling. Beyond, let's say 2-3 cm, water depth is of no importance to call production in our species. We also understand that it would be hard to imagine what sort of physics would be underlying such effect. We have tried to make clear that all the interactions we are interested in take place at the boundary between being completely afloat and touching the ground with all body parts. We do however like to point out that we did manipulate the water depth, not floating or amount of air taken up by the frogs. Therefore, we can only make clear statements about the relationship between water depth and calling, but we will make clear that this relationship is probably an indirect one caused by the amount that a frog is touching the ground.

L98: I pity the authors (and readers) for having to deal with this journal's cumbersome style of banishing the methods to the end of the paper. This is a real curse for the flow of reading and understanding one's work!

L100ff: I don't fully understand how these numbers are supposed to be read and how they add up. Relative to what were the water bodies of males that modified their call site "shallow" – as far as I understood they actually *deepened* their site (9.4 vs 1.39 cm)?! What was 1.55 +/- 0.44 cm – was this the average difference between natural and modified puddles, or the increase in depth frogs in the very puddles the frogs deepened?

See our reply to reviewer 1. Part of the confusion was caused by a typo, for which we apologize. We have rephrased this section to make clear what we actually measured.

L103: please rephrase – the frogs did not create deeper puddles as a consequence *by* calling from shallow water bodies, as your wording suggests *sensu stricto*.

Correct, thanks for catching this error!

L107: Please reconsider and rephrase accordingly - as following from simple physical laws, floating is rather a strict function of water depth, and size and buoyancy of the frogs, I don't understand why this is considered an "ability of males", nor why this even requires testing. The actually interesting information is rather, which water depth is required to ensure floating for the average Tungara, and the passage should be rephrased and presented accordingly.

For our species, floating is not directly derived from simple physical laws. It is probably the result of the amount of air taken up by the frog in relation to its body size and as such it does reflect an animal's ability (behaviour). But we have rephrased it nonetheless.

L109ff: And how about deep water? I miss the corresponding information!?

See also response to reviewer 1. We have added the information accordingly.

L159: In my opinion the focus on “water depth”, also in the discussion, gives the entire paper a twist in the wrong direction. The actual effect on calling activity and characteristics comes from floating vs. not-floating which only then is (mostly) a function of water depth (and to a lesser extent to animal size and density).

See our general reply above.

L251: Change “our species” against the scientific or vernacular name

We have rephrased this part

L252: Isn't this only the case for one of the signals modalities (water ripples), whereas in the other modality (airborne sound) actually the conspicuousness is (still) increased – as it is intended to increase the attractiveness for females?! Cf. L55 where you say that “signal components” can be used by predators for detection. Or are tungaras safe from airborne acoustic eavesdropping?

Correct, we have rephrased this in more general terms.

L259: Can you elaborate a little on the specifics of “concrete structures associated with the drainage system”, especially the water levels tungaras can expect there?

We have removed this part after comments from reviewer 1

L264: Cf. L79 – at least please be consistent with your nomenclature!

done!

L270, check throughout: “10–20” – use – (dash) not – hyphen to indicate ranges

done!

L279: Is toe-clipping strictly necessary for individual recognition of Tungara frogs or would there have been other feasible methods like pattern matching? Was individual identification for this study the only purpose of this toe clipping, or were tissue samples collected also for other purposes? Was toe-clipping only for marking specifically mentioned/allowed in your experimental permits? Can you present evidence of the effects of toe-clipping in this species? Please explain.

Recognition is mainly necessary to avoid retesting the same individual. Toe-clipping is part of a generally procedure to ID frogs used by different labs and to collect tissue samples for long-term genetic studies. This method has been approved by the American Society of ichthyologists and herpetologist

<http://www.asih.org/sites/default/files/documents/resources/guidelinesherpsresearch2004.pdf>

L298: I assume water depths of 0.25 – 2.0 cm ?!+

Correct!

L300: Please specify the *acoustic* characteristics of the speaker and also give information on the playback device; here and elsewhere, give complete maker information (type, company,

city, (state), country).

We have added more information to the methods

L301: does “synthetic” refer to a signal that was digitally created from scratch, or one that was composed from recorded parts. Please specify the characteristics of the signal in more detail and how/why they were chosen (e.g. population average, etc.)? Please also give specifications and origin of the chorus stimulus.

The stimulus is based on the population average and constructed from scratch. We refer to the original research that used it for the first time for further details

L310: please specify the dimensions of the room and how far the speakers were away from the tested female.

We have added this

L311ff: Your wording is confusing to understand the actual experimental setup and its rationale – you recorded male frogs at 4 water depths, but used only recordings from 3 depths in the playbacks (why?) with females, but categorize them as only shallow (2 different depths) or deep (only 1 depth) (why?). This is confusing and needs clearer writing.

We had few males calling in both the 0.25 and 0.5 cm treatment and as a result decided to use the best recording (highest call rate, etc.) from either of these treatments. We have clarified this in the text.

L312: Your wording “minimum of 6 calls” suggests that you used stimuli with a varying number of calls per time? What was the actual range/average of the call rate? Why did you not manipulate the stimuli to standardize the call rate across stimulus conditions? In the introduction you did not explicitly suggest call rate as a potential function of water depth – did you actually see/treat call rate as a part of the experimental condition, or could it be an overlooked confounding effect on female preference? If you don’t think so, please lay out your rationale.

Water depth treatment had a clear effect on call rate, just as the other acoustic features we measured (see also table 1). For the preference tests we standardised call rate (selecting the call with the highest complexity from a recording with a minimum of 6 calls). We have clarified this now.

After having read the entire paper it becomes clear, that you actually do see also call rate as a function of floating vs. non-floating (water depth) – this should be made clear much earlier in the manuscript

L318: There is missing a lot of information here: How long was the overall trial length? For how long was the acoustic stimulus presented – or did you simply directly use the 1 min recordings from the male trials? If you just used the 1 min recording, does this mean there was 1 min of silence before you lifted the funnel – or did you play the 1 min recording twice? Did you continue playing the stimulus after the funnel was lifted, or did you stop the playback and the females were still deciding on one side although there was no stimulus anymore? Did you stop the trials after a defined time when females did not move? If so, how many did not move? Did all females immediately and unambiguously decide for a speaker or did you have a criterion in place to score decisions?

We had added extra information on the preference tests

L321: I suggest the heading “Data collection in the field”

We also collected data in the lab

L326: the diameter of the dug-out puddle is rather a result, no?

Correct, we moved it to the result section

L327: Unclear reflexive wording – does “these puddles” here refer to the larger, natural body of water, or to the excavation by the frogs?

Unclear wording - Naturally, the “depth at the edge of a puddle” converges against 0, unless you have a straight wall perpendicular to the water surface. Also I don’t understand, how you can “measure depth...in a predetermined direction” – do you want to say that you measured the depth of (the natural) puddles at a defined (how much?) distance from the edge, in a defined (which?) direction? Please rephrase for clarity.

We have rephrased this part

L330: I suggest to integrate the specification of the technical equipment in the experimental setup into the according paragraphs (L285ff)

We prefer to leave it here

L331ff: complete maker information please

We have provided the information

L334: “...at an angle of 45° to the water surface, and at a distance...”

Thanks!

L343: There it is - this is complete maker information for the first time!

!

L347: “16 bit sampling depth”

changed it

L349-350: delete one “either”

done!

L356ff: Rephrase for clarity – “the videos” - as it reads now, it only becomes clear at the end of the paragraph, when you mention the cage, that this is about filming of calling males in the laboratory, and not about the high speed filming under natural conditions.

We have clarified this

L370: Please give a proper reference for ImageJ as per the suggestion of its authors:

<https://imagej.nih.gov/ij/docs/faqs.html>

L383: In the context of this study, I do not understand where and why normalization was needed – please explain. What do you mean by “for visual purposes”? I can only assume that

this refers to L483 – the envelopes and spectrograms of normalized calls and vibrations? If this is the case, leave it out here – an explanation in the figures legend is sufficient.

We have changed this accordingly

L386/397: “...up to three calls from each male...” suggests, that you averaged amplitude measurements within males, or that “individual male” was a random factor in your model, but this is not clearly stated.

We have clarified this

L390-391: cf. L270, please be consistent with the format of reporting ranges

L396: Please give a proper reference for R as per the authors suggestion: <https://cran.r-project.org/doc/FAQ/R-FAQ.html#Citing-R>

L457: The somewhat hidden logarithmic x-axis (doubling of the distance with every tick) is misleading in suggesting a strong and linear relationship between water depth and call parameters; actually the effect is far less than suggested and it is not linear but rather logarithmic/asymptotic. I rather see the characteristics of a phase-transition/tipping point from non-floating to floating (which then depends on water depth) then a direct influence of water depth on the call parameters. This is especially the case for “call complexity”. Returning to L395 - how was water depth coded for the linear mixed model – on an ordinal or on a linear scale?

It was on a linear scale, we have removed the lines from the figures to avoid misleading the readers.

REVIEWERS' COMMENTS:

Reviewer #1 (Remarks to the Author):

My concerns were relatively minor and have been addressed in the revisions. I appreciate the detailed comments from Reviewer #2 and believe that the manuscript was improved by addressing these comments.

Reviewer #2 (Remarks to the Author):

I am more than happy with the way the authors have improved their manuscript following the other reviewers and my suggestions. I agree with the authors that, instead of a total reframing of the manuscript, their tweaks and rewordings are sufficient to remove my (and hopefully also other readers) confusion regarding the role of floating on the calling frogs. I see the manuscript ready for publication, pending a few minor changes and suggestions on style and writing (see below).

Kind regards

Max Ringler (rev #2 of the original submission)

--

L82 (previously L79): I did not even refer to the ongoing taxonomical/nomenclatural skirmishes...my remark was just about punctuation and italicization: L22: (*Physalaemus* = *Engystomops pustulosus*) vs. L82: (*Physalaemus* [= *Engystomops*] *pustulosus*) vs. L283: (*Physalaemus* [=*Engystomops*] *pustulosus*) (here *Physalaemus* not in italics) which should be consistent

L83: reword (remove the structure with parenthesis): ...in shallow water bodies, typically a few centimetres in depth, that are...

L119: sample size is missing

L128: reword: ...a behaviour that in the field is associated...

L136, 139, 146, 147, elsewhere: Please check carefully throughout the entire manuscript and consider specifically mentioning "water level" instead of just "level", at least at the first instance in a sentence, where there is a chance for ambiguity or for intuitive confusion. E.g "called at the lowest level" is prone to misunderstanding, as it is first intuitively associated with SPL/amplitude levels, even though it *should* be somehow clear from the context.

L160: should read "...0.25 cm depth..."

L283: italics for *Physalaemus*

L326-326: Apparently there is something wrong with the placement of the maker information, which should follow directly after the instrument, not after its configuration

L320 vs. L333: In the maker information, either always spell out states, or give the abbreviation (preferred), but please be consistent

L337: "...called in the 0.25..."

L339: "...the 2.0 cm treatment..."

L340: please explain "maximum complexity" here – I assume in the Tungara-world this means "highest number of chucks", but this needs to be explained to all readers.

L359: If my understanding is right this time, you actually measured the water depth in a part of the puddle that had not been modified by the frog?! If this is correct, please try to reword to explicitly say so.

L364: Again, please rather keep maker information with the instrument, followed by its settings/characteristics

L365: The Avisoft A/D interface is not made by G.R.A.S. I assume

L366: As this is somehow complex/confusing here, maybe something like ";this and all other G.R.A.S. equipment made by..." after the first mention?

L367: As before in L 334 of the original submission, this is still not fixed: most certainly the microphone was pointing downwards to the frog at an angle of $\sim 45^\circ$ from the horizontal/towards the water surface – this just should be stated explicitly for clarity; further, was the 50 cm the horizontal distance, or the recording distance to the microphone? At what height was the microphone placed? (if it was 50 cm horizontal distance, then obviously the microphone was at 50 cm height when pointing down at an angle of 45°).

L377: This is how I would prefer the maker information, with the settings/specifications at the end. Sorry, you must be pretty annoyed already by me pointing this out over and over again. I simply struggled more than once in the past to find more information about equipment used in other studies, that I wanted to use/try myself. And this is easy and straightforward to fix, so hopefully not too annoying...

L379: *blushing*...and here it is missing again for the Marantz recorder

L428: I am most certain the authors of lme4 would also appreciate proper citation:

Douglas Bates, Martin Maechler, Ben Bolker, Steve Walker (2015). Fitting Linear Mixed-Effects Models Using lme4. Journal of Statistical Software, 67(1), 1-48.<doi:10.18637/jss.v067.i01>

Download available at:

<https://cran.r-project.org/web/packages/lme4>

L441: either "of the fixed effect (or effects)" or "of fixed effects"

Figure 1: I believe you misunderstood my point – it is the X-axis that *is* logarithmic in the sense, that from one box of the boxplot to the next, which are spaced at equal distances, the water depth always doubles – 0.25-0.5-1-2 cm ... to me this is a logarithmic scale.

reply to reviewers:

REVIEWERS' COMMENTS:

Reviewer #1 (Remarks to the Author):

My concerns were relatively minor and have been addressed in the revisions. I appreciate the detailed comments from Reviewer #2 and believe that the manuscript was improved by addressing these comments.

Thanks!

Reviewer #2 (Remarks to the Author):

I am more than happy with the way the authors have improved their manuscript following the other reviewers and my suggestions. I agree with the authors that, instead of a total reframing of the manuscript, their tweaks and rewordings are sufficient to remove my (and hopefully also other readers) confusion regarding the role of floating on the calling frogs. I see the manuscript ready for publication, pending a few minor changes and suggestions on style and writing (see below).

Kind regards

Max Ringler (rev #2 of the original submission)

--

L82 (previously L79): I did not even refer to the ongoing taxonomical/nomenclatural skirmishes...my remark was just about punctuation and italicization: L22: (*Physalaemus* = *Engystomops pustulosus*) vs. L82: (*Physalaemus* [= *Engystomops*] *pustulosus*) vs. L283: (*Physalaemus* [=*Engystomops*] *pustulosus*) (here *Physalaemus* not in italics) which should be consistent

We italicized it all

L83: reword (remove the structure with parenthesis): ...in shallow water bodies, typically a few centimetres in depth, that are...

we have rephrased this

L119: sample size is missing

we have added it

L128: reword: ...a behaviour that in the field is associated...

done!

L136, 139, 146, 147, elsewhere: Please check carefully throughout the entire manuscript and consider specifically mentioning “water level” instead of just “level”, at least at the first instance in a sentence, where there is a chance for ambiguity or for intuitive confusion. E.g “called at the lowest level” is prone to misunderstanding, as it is first intuitively associated with SPL/amplitude levels, even though it *should* be somehow clear from the context.

done!

L160: should read “...0.25 cm depth...”

changed!

L283: italics for Physalaemus

done

L326-326: Apparently there is something wrong with the placement of the maker information, which should follow directly after the instrument, not after its configuration

we have change it

L320 vs. L333: In the maker information, either always spell out states, or give the abbreviation (preferred), but please be consistent

we abbreviated states

L337: “...called in the 0.25...”

L339: “...the 2.0 cm treatment...” changed

L340: please explain “maximum complexity” here – I assume in the Tungara-world this means “highest number of chucks”, but this needs to be explained to all readers.

we have explained this.

L359: If my understanding is right this time, you actually measured the water depth in a part of the puddle that had not been modified by the frog?! If this is correct, please try to reword to explicitly say so.

we have specified this

L364: Again, please rather keep maker information with the instrument, followed by its settings/characteristics

L365: The Avisoft A/D interface is not made by G.R.A.S. I assume

but the mic is.

L366: As this is somehow complex/confusing here, maybe something like “;this and all other G.R.A.S. equipment made by...” after the first mention?

L367: As before in L 334 of the original submission, this is still not fixed: most certainly the microphone was pointing downwards to the frog at an angle of ~45° from the horizontal/towards the water surface – this just should be stated explicitly for clarity; further, was the 50 cm the horizontal distance, or the recording distance to the microphone? At what height was the microphone placed? (if it was 50 cm horizontal distance, then obviously the microphone was at 50 cm height when pointing down at an angle of 45°).

the microphone was placed on a tripod, pointing to the surface at 45 degrees. The distance was kept at 45 cm between tip of the mic and the position of the frog. This should be clear from the text.

L377: This is how I would prefer the maker information, with the settings/specifications at the end. Sorry, you must be pretty annoyed already by me pointing this out over and over again. I simply struggled more than once in the past to find more information about equipment used in other studies, that I wanted to use/try myself. And this is easy and straightforward to fix, so hopefully not too annoying...

we've tried to fix this as much as possible.

L379: *blushing*...and here it is missing again for the Marantz recorder

L428: I am most certain the authors of lme4 would also appreciate proper citation:

Douglas Bates, Martin Maechler, Ben Bolker, Steve Walker (2015). Fitting Linear Mixed-Effects Models Using lme4. Journal of Statistical Software, 67(1), 1-48.<doi:10.18637/jss.v067.i01>

we already cite the r program

Download available at:

<https://cran.r-project.org/web/packages/lme4>

L441: either “of the fixed effect (or effects)” or “of fixed effects”

changed it

Figure 1: I believe you misunderstood my point – it is the X-axis that *is* logarithmic in the sense, that from one box of the boxplot to the next, which are spaced at equal distances, the water depth always doubles – 0.25-0.5-1-2 cm ... to me this is a logarithmic scale.

correct, which is why we removed the line in the previous version.